# Dissecting the sharp response of a canonical developmental enhancer reveals multiple sources of cooperativity

Jeehae Park, Javier Estrada[†], Gemma Johnson, Ben J Vincent[‡], Chiara Ricci-Tam, Meghan DJ Bragdon, Yekaterina Shulgina, Anna Cha, Zeba Wunderlich[§], Jeremy Gunawardena, Angela H DePace*

Department of Systems Biology, Harvard Medical School, Boston, United States

*For correspondence:
Angela_DePace@hms.harvard.edu

Present address: [†]Novartis Institutes for Biomedical Research, Cambridge, United States; [‡]Department of Biological Sciences, University of Pittsburgh, Pittsburgh, United States; [§]Department of Developmental and Cell Biology, University of California Irvine, Irvine, United States

Competing interests: The authors declare that no competing interests exist.

**Abstract** Developmental enhancers integrate graded concentrations of transcription factors (TFs) to create sharp gene expression boundaries. Here we examine the hunchback P2 (HbP2) enhancer which drives a sharp expression pattern in the Drosophila blastoderm embryo in response to the transcriptional activator Bicoid (Bcd). We systematically interrogate cis and trans factors that influence the shape and position of expression driven by HbP2, and find that the prevailing model, based on pairwise cooperative binding of Bcd to HbP2 is not adequate. We demonstrate that other proteins, such as pioneer factors, Mediator and histone modifiers influence the shape and position of the HbP2 expression pattern. Comparing our results to theory reveals how higher-order cooperativity and energy expenditure impact boundary location and sharpness. Our results emphasize that the bacterial view of transcription regulation, where pairwise interactions between regulatory proteins dominate, must be reexamined in animals, where multiple molecular mechanisms collaborate to shape the gene regulatory function.
DOI: https://doi.org/10.7554/eLife.41266.001

## Introduction

During development, multicellular organisms control cell differentiation by expressing genes in intricate patterns in space and time. These patterns are achieved by interactions between regulatory proteins and DNA, which convert incoming signals into transcription of mRNA; we refer to the quantitative relationship between the concentrations of transcription factors and the rate of mRNA, or protein, expression in steady state as the Gene Regulatory Function (GRF). Many GRFs for developmental genes are highly non-linear; they convert graded inputs to sharp, step-like outputs. A key mechanistic question is therefore how a sharp GRF is encoded in the regulatory DNA sequence and the interactions of proteins that regulate it.

In animals, gene expression patterns are controlled by enhancers, regulatory DNA comprised of multiple binding sites for transcription factors (TFs) (*Long et al., 2016*; *Spitz and Furlong, 2012*). The canonical model of how an enhancer gives rise to a non-linear GRF focuses on pairwise cooperative TF binding to these sites (*De Val et al., 2008*; *Jolma et al., 2015*; *Panne et al., 2007*; *Rodda et al., 2005*; *Struhl, 2001*), an idea that first emerged to explain gene regulation in phage lambda (*Johnson et al., 1979*) and has since been widely applied to other systems including eukaryotes. For example, the *Drosophila melanogaster* Hunchback P2 enhancer (HbP2) drives a highly nonlinear GRF in response to the transcriptional activator Bicoid (Bcd) in the early embryo; it has long been thought that the nonlinearity arises from pairwise cooperative binding of Bcd to six DNA binding sites in the enhancer (*Burz et al., 1998*; *Driever et al., 1989*; *Ma et al., 1996*; *Struhl et al., 1989*). Eukaryotic TFs can indeed influence each other's binding through direct protein-protein interactions (for Bcd examples include (*Burz et al., 1998*; *Burz and Hanes, 2001*; *Lebrecht et al., 2005*),

**eLife digest** Building an organism from scratch requires genes to be switched on or off at precisely the right time, in the right place, and at the right level. Enhancers are stretches of DNA that work as switches to turn on target genes. For instance, in the front part of fruit fly embryos, the P2 enhancer switches on a gene called *Hunchback*, which is crucial for development.

A number of molecular actors, including proteins called transcription factors, work together to turn on genes by interacting with enhancers. Genes like *Hunchback* can turn on suddenly, even though they are controlled by transcription factors whose levels are changing gradually: in other words, if *Hunchback* were controlled by a light switch with a dimmer, the light would suddenly come on as the dimmer was gradually moved up. For enhancers, the question is how transcription factors interact with DNA to convert a gradual input into an abrupt, sharp switch. A commonly accepted view is that *Hunchback* is turned on when molecules of a transcription factor called Bicoid help each other to bind to multiple binding sites on the P2 enhancer.

Park et al. investigated this mechanism by examining how the *Hunchback* gene responded to changes in the sequence of the P2 enhancer, and to changes in the levels of regulatory proteins that bind to it. The resulting observations were then compared to mathematical models that simulate turning on *Hunchback* under different conditions. The experiments revealed that, in fact, switching on *Hunchback* requires more than Bicoid proteins helping each other to bind on the P2 enhancers. Molecules other than Bicoid were also needed, and the cell also potentially had to burn energy.

Variations in the sequence of enhancers are linked to evolution of new species but also to problems in development or even diseases such as cancer. Understanding precisely how these sequences turn on genes will give us insight into which types of changes are important for disease and evolution.

DOI: https://doi.org/10.7554/eLife.41266.002

and this observation led to substantial efforts to find a 'cis-regulatory code' where the position, orientation and affinity of TF binding sites can predict the output of a given enhancer (*Yáñez-Cuna et al., 2013*). However, such a code has remained elusive (*Catarino and Stark, 2018*), indicating that there are likely additional molecular mechanisms contributing to non-linear GRF formation.

In addition to direct protein-protein interactions between TFs, there are many ways that TFs can cooperate indirectly, especially in eukaryotes (*Gertz et al., 2009*; *Spitz and Furlong, 2012*). The eukaryotic transcriptional machinery involves not only TFs, but also chromatin remodeling machinery, and cofactors that relay information about TF binding to the basal transcriptional machinery (*Allen and Taatjes, 2015*; *Hargreaves and Crabtree, 2011*; *Levine, 2010*; *Wang et al., 2013*; *Zentner and Henikoff, 2013*). Thus cooperativity between TFs can potentially arise indirectly through any mechanism that facilitates subsequent TF binding, or through synergistic activation of the basal transcriptional machinery. Indirect mechanisms that can affect TF binding include collaborative cooperativity, where TFs each affect nucleosome binding or modification state (*Mirny, 2010*; *Voss et al., 2011*), TFs co-binding to a co-factor complex (*Allen and Taatjes, 2015*; *Borggrefe and Yue, 2011*; *Wang et al., 2013*), TFs altering the topology of DNA (*Courey, 2001*) or TFs increasing the effective local concentration of other proteins (*Landman et al., 2017*). Indirect mechanisms that do not rely on facilitating TF binding include synergistic activation of the basal transcriptional machinery through multiple allosteric interactions (*Nussinov et al., 2013*), or synergy through activation of multiple steps in the transcription cycle (*Coulon et al., 2013*; *Duarte et al., 2016*; *Govind et al., 2005*; *Herschlag and Johnson, 1993*; *Scholes et al., 2017*).

We recently used mathematical theory, grounded in molecular biophysics, to explore the mechanisms underlying nonlinear GRFs generated by a single activating TF, inspired by the example of HbP2 activation by Bcd (*Estrada et al., 2016b*). GRFs are often characterized by fitting experimental data to a Hill function, which parameterizes the shape of a GRF in terms of three parameters (below) but does not give any mechanistic or biophysical insight into how the GRF arises (*Engel, 2012*; *Weiss, 1997*). The GRF of early anterior Hunchback expression with respect to Bcd fits a Hill function with coefficient 5–6. It is often informally assumed that this can be explained in terms of pairwise cooperative binding of Bcd to six sites in HbP2, with the Hill coefficient matching the number of sites

(*Burz et al., 1998*; *Driever et al., 1989*; *Gregor et al., 2007*; *Lopes et al., 2005*; *Ma et al., 1996*). We used our mathematical theory to prove that this assumption is unfounded, providing further evidence that additional mechanisms are required to fully explain the response of HbP2 (*Xu et al., 2014*; *Singh et al., 2005*; *Fu et al., 2004*). We showed that if there are $n$ binding sites for a transcriptional activator, the GRF can reach a Hill coefficient of nearly $n$ only through two non-exclusive mechanisms: either there is 'higher-order' cooperativity, through which binding of a TF to one site can be influenced by multiple other sites, or there is energy expenditure which maintains the regulatory system away from thermodynamic equilibrium.

In this study, we experimentally probe the mechanisms underlying the shape of the HbP2 GRF using quantitative experiments and compare the results to our previously developed theory. We created a number of variants of HbP2, where we changed the number of Bcd binding sites and the background sequence of HbP2; we measured the resulting GRFs using quantitative imaging of reporter constructs in embryos. A synthetic version of HbP2 containing only Bcd binding sites indicates that other TFs play a role in determining the anterior/posterior position of the GRF. We perturbed the concentration of multiple transcriptional cofactors using RNAi and found that cofactors can also affect the shape of the HbP2 GRF, both in terms of the anterior/posterior position of expression and the steepness of the curve. Finally, we compared our experimental results to theoretical predictions for HbP2 and its variants; this comparison confirms our previous conclusion that higher cooperativity and/or energy expenditure are required to achieve the sharp response of HbP2 to Bcd. Together, our results demonstrate that the shape of the non-linear HbP2 GRF is determined by multiple underlying molecular mechanisms which can be regulated independently and collectively by distinct inputs.

## Results

### The HbP2 reporter system

Our study focuses on a canonical developmental enhancer, HbP2, that drives expression of hunchback (hb) in the *Drosophila melanogaster* blastoderm embryo. The anterior expression of hb is set by three enhancers: HbP2, which is promoter-proximal, a distal shadow enhancer and a distal stripe enhancer (*Figure 1*). HbP2 drives early expression of hb at nuclear cycle 14, while the shadow enhancer and stripe enhancer contribute to expression at later times (*Perry et al., 2012*; *Perry et al., 2011*). Many studies have correlated in vivo hb expression (using either hb mRNA or protein) to HbP2 sequence features without accounting for these other enhancers (*Gregor et al., 2007*; *Duarte et al., 2016*; *Lopes et al., 2012*); this approach is only valid when analysing early timepoints when HbP2 is active and the others are not, and even then may suffer from errors. To control for this complexity, we isolated HbP2 in a reporter construct consisting of HbP2 and the native hunchback promoter driving expression of lacZ (hereafter called WTHbP2). This construct is the basis for our permuted enhancers, including Bcd binding site deletions and synthetic hbP2 constructs (SynHbP2 plus variants thereof). We inserted all reporter constructs into the attP2 landing site on chromosome 3LT via PhiC31 integrase-mediated transgenesis (see Materials and Methods).

To visualize the activity of our reporter constructs via expression of LacZ, we fluorescently stained fixed embryos using in situ hybridization against LacZ (see Materials and Methods). We measured the Bcd expression profile using fluorescent immunostaining, both separately and in embryos costained for LacZ mRNA (see Materials and Methods). We imaged entire embryos using 2-photon microscopy and parsed image stacks into cellular resolution pointclouds using a semi-automated pipeline (see (*Fowlkes et al., 2011*; *Fowlkes et al., 2008*) and Materials and Methods). Full 3D images of embryos are particularly useful for alignment, which reduces error due to dorsal ventral variation in the hb expression pattern. We gathered data from the earliest stages of nuclear cycle 14 when HbP2 is active. In subsequent figures, we present a subset of this full 3D dataset for simplicity; we show a lateral trace along the anterior-posterior (AP) axis (*Figure 1A*). Full cellular resolution data are available for download from Figshare (https://doi.org/10.6084/m9.figshare.8235491.v1 [1]).

To analyze the relationship between Bcd concentration and transcription driven by HbP2, we extract the gene regulatory function (GRF) relating these two quantities. The Bcd gradient is highly reproducible from embryo to embryo (*Gregor et al., 2007*); we therefore plot the average Bcd protein profile from six embryos against the LacZ expression profile from individual embryos stained for

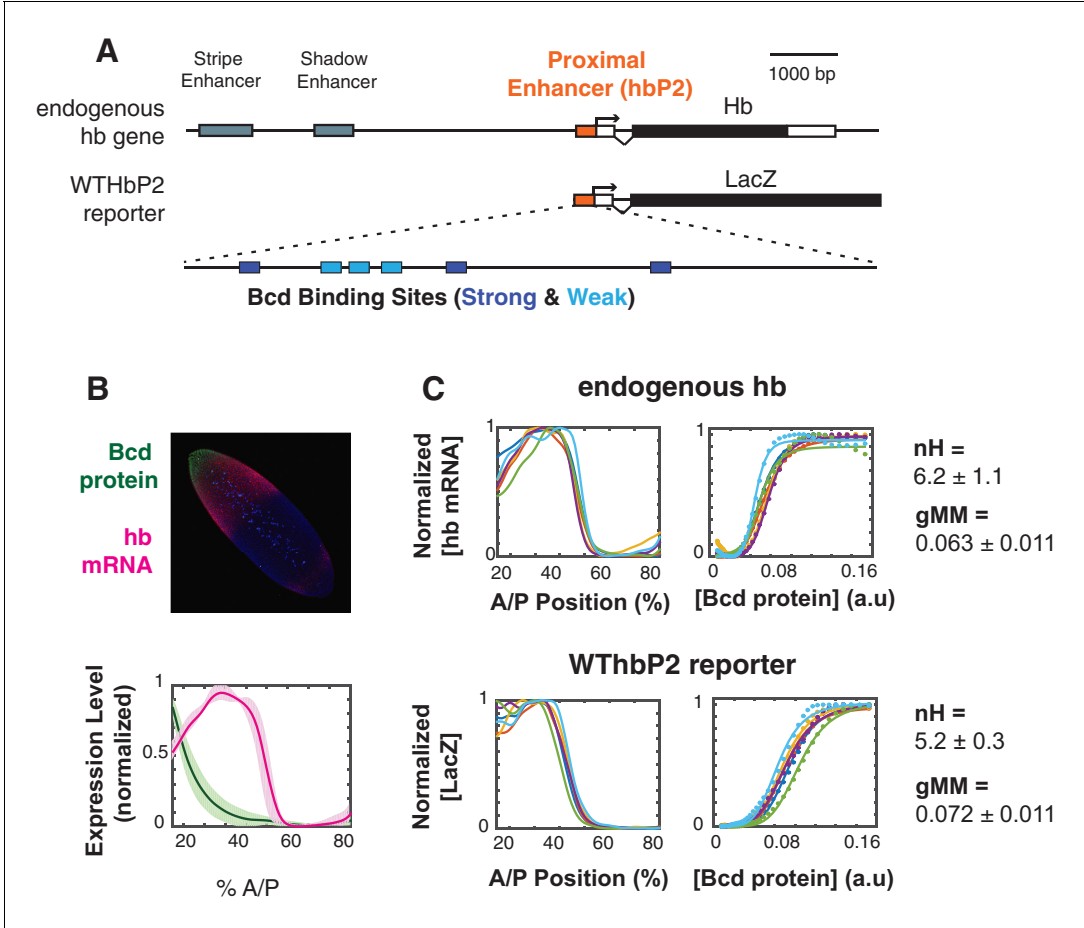

**Figure 1.** The WTHbP2 enhancer directs a sharp expression pattern in the Drosophila blastoderm embryo. (**A**) Schematics of the hunchback (hb) gene locus showing the three enhancers active in the blastoderm (top) and our reporter construct designed to express lacZ mRNA from the proximal enhancer (HbP2) and hb native promoter. HbP2 has six foot-printed Bicoid binding sites (bottom). (**B**) (top) A *Drosophila melanogaster* blastoderm embryo at nuclear cycle 14 stained for Bicoid (Bcd) protein and hb mRNA; image is a maximum projection of z-stack. (bottom) The average Bcd protein and hb expression profiles along anterior-posterior (AP) axis measured at midsagittal plane; data from six embryos. Average is represented by the thick line, standard deviation is shown by the shaded area. (**C**) GRFs for hb mRNA and WTHbP2-reporter expression profiles. Expression of hb or LacZ along the anterior posterior axis is plotted on the left; hb or LacZ expression relative to Bcd concentration is shown on the right. Colors represent individual embryos. Lines on the left are extracted expression traces. On the right, each dot is a measured mRNA level and lines represent fits to the Hill function. From these data, we computed the Hill coefficient (nH), which reflects the shape of the curve, and generalised Michaelis-Menten constant (gMM) which reflects the location of the expression boundary along the anterior/posterior axis.

DOI: https://doi.org/10.7554/eLife.41266.003

LacZ mRNA driven by the HbP2 reporters. We chose to analyze individual reporter output against the average Bcd profile for two reasons. First, by obtaining an average Bcd profile once and using it as a standard, our embryos can be stained either for protein or RNA, but not both; incompatibilities between in situ hybridization (to detect LacZ) and immunostaining (to detect Bcd protein) lead to noisier data than staining for either alone. Second, we can detect if our perturbations affect variability in expression pattern between individual embryos, which would otherwise be masked in average to average fitting which is commonly used (*Gregor et al., 2007*; *Lopes et al., 2012*).

For convenience of analysis, we will describe the shape of our GRFs by fitting lateral traces to a Hill function of the form,

$$E = Emax \, [Bcd]^{nH} / \left( Ka^{nH} + [Bcd]^{nH} \right)$$

Here, E is the hb mRNA expression level in steady state, in arbitrary units, with Emax being its maximal value. The Hill coefficient, nH, is a measure of how sharply the GRF changes, while the

parameter Ka gives the concentration at which the expression level becomes half-maximal and determines the location of the GRF on the anterior-posterior (AP) axis of the embryo. We will refer to Ka as the 'generalised Michaelis-Menten constant' (gMM constant), as it reduces to the well-known Michaelis-Menten constant when the GRF becomes a hyperbolic function with a Hill coefficient of 1. As noted above, the Hill function gives little biophysical insight but it offers a widely-used measure of shape. We will later interpret the shape of the GRF in terms of the mathematical theory that we introduced previously, from which we will draw more mechanistic conclusions.

We measure a Hill coefficient for the endogenous hb mRNA expression pattern of 6.2 +/- 1.1, which corresponds with other reports in the literature (*Gregor et al., 2007*; *He et al., 2011*; *Lopes et al., 2012*). For WTHbP2, the Hill coefficient is 5.2+/- 0.3, indicating that HbP2 enhancer alone can drive sharp expression comparable to expression from native hb locus. The boundary position for the HbP2 pattern (gMM = 0.072 ± 0.011) is also comparable to that of the endogenous hb (gMM = 0.063 ±0.011).

## HbP2 variants do not adhere to the classical hill function model

A common assumption based on Hill fits to the endogenous Hb mRNA pattern is that Bcd binds cooperatively to 6 Bcd binding sites in HbP2 and that this gives rise to a Hill coefficient of ~6; more generally this model states that cooperative binding at *n* sites gives rise to a Hill function with coefficient nH = *n*. A concrete test of this hypothesis is to measure the GRF when Bcd binding sites are removed from WTHbP2 where this 'classical Hill function model' predicts two outcomes. First, removing all Bcd binding sites from HbP2 should eliminate expression. Second, deleting individual Bcd binding sites should lead to a progressive, integral decrease in the Hill coefficient. We tested the classical Hill function model by mutating individual Bcd binding sites in the WTHbP2 reporter and measuring LacZ expression. We deleted each Bcd binding site in series starting from the distal end relative to the promoter (*Figure 2A*).

First, eliminating all 6 Bcd binding sites does not abolish expression. This indicates that other binding sites, either for Bcd or other TFs, contribute to HbP2 expression, which has been previously argued in the literature (*Chen et al., 2012*; *Holloway and Spirov, 2015*; *Liu and Ma, 2013*; *Mito et al., 2006*; *Xu et al., 2014*). We analyzed the sequence of HbP2 and hb native promoter and found two predicted binding sites for Bcd. Removing these additional sites further reduced gene expression level, however it did not completely abolish it (*Figure 2—figure supplement 1*). Second, while the Hill coefficient does decrease as Bcd sites are removed, it only decreases by ~0.5 for each site removed. Removing Bcd binding sites also decreases the gMM constant and shifts the expression of HbP2 toward the anterior. Together, these results indicate that the assumption that Hill coefficient follows a one-to-one relationship with the number of binding sites in the enhancer is wrong, as we would have predicted from our biophysical model.

A common question is whether individual Bcd binding sites contribute differently to the WTHbP2 GRF. Of the six Bicoid binding sites, three are strong affinity sites (sTFBSs) and three weak affinity sites (wTFBSs). wTFBSs are clustered together at the center of WTHbP2 while sTFBS are located on the flanks with varying distance to the nearest Bcd binding site (13 bp to 106 bp, center to center). The role of these individual sites has been investigated in terms of pairwise interactions between Bcd molecules in vitro (*Ma et al., 1996*) and in yeast (*Burz and Hanes, 2001*), where binding is influenced by the location and orientation of the sites. The role of individual binding sites in embryos has been explored, but without quantitative imaging (*Driever et al., 1989*), making the effects difficult to discern.

We also did limited experiments to test whether the affinity or location of the weak and strong Bcd binding sites influences the WTHbP2 GRF in our reporter construct. To compare the effect of weak and strong Bcd binding sites, we deleted single binding sites with different affinities (*Figure 2C*). In all cases, removing a Bcd binding site decreased the Hill coefficient by ~0.5 indicating that this parameter was not disproportionately affected by any particular site. The gMM constant also decreased sequentially upon TFBS removal without distinctive dependence on any particular single site. To test whether the cluster of weak Bcd binding sites has a disproportionate effect on sharpness, we deleted sets of two and three binding sites, either all in the central cluster or not. We found removing Bcd binding sites from the central cluster differentially affects both the gMM constant and the Hill coefficient, with the central cluster having a moderately larger effect on the gMM but the differences are within error (*Figure 2C*). Together, these results indicate that all six binding

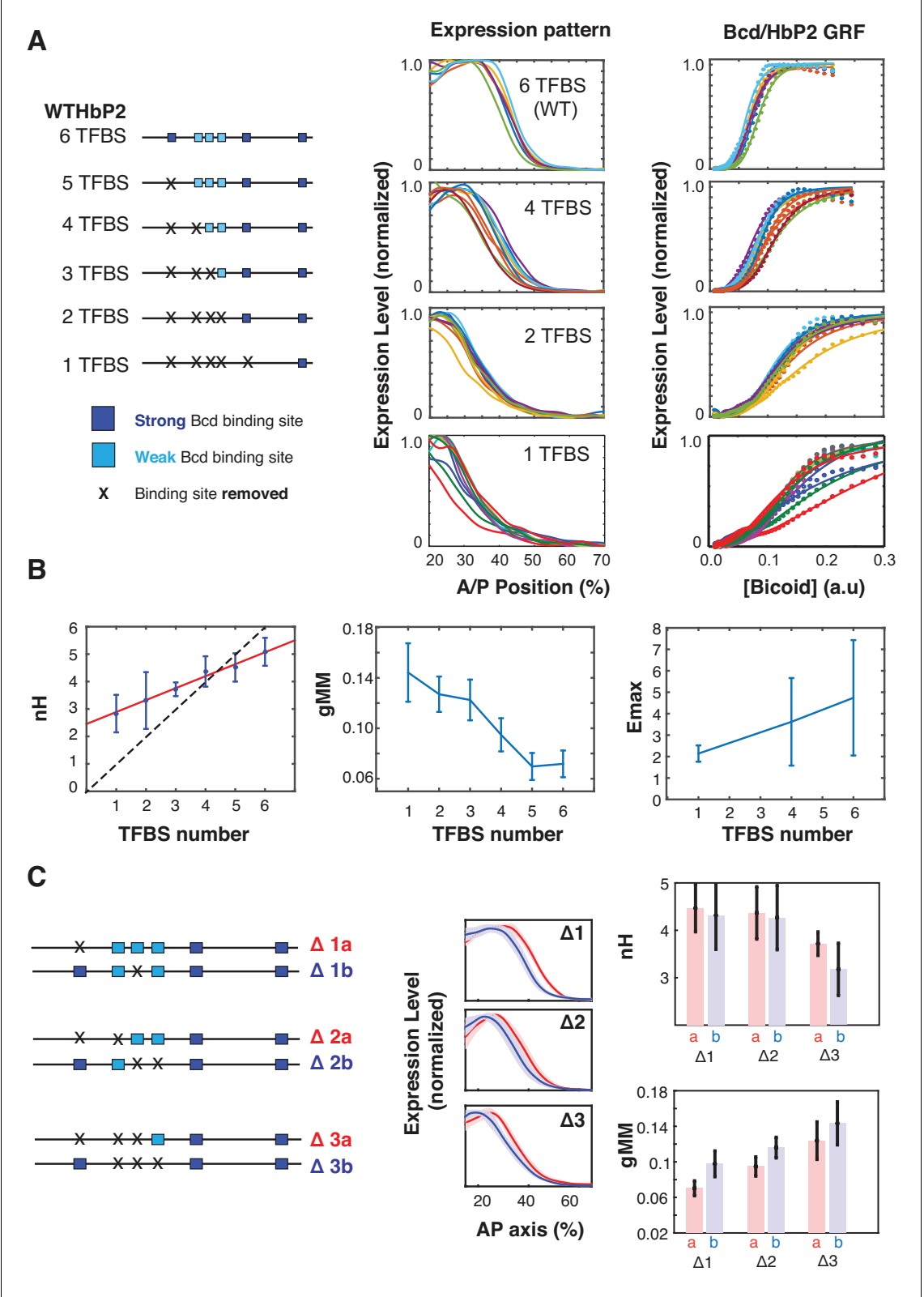

**Figure 2.** Sequence variants of HbP2 contradict the classical Hill function model. (**A**) (left) Schematics of hbP2-reporter constructs with Bcd binding sites sequentially removed by mutation to preserve the length of the construct. (center) mRNA expression profiles obtained from embryos that carry transgenic reporter constructs shown on the left. (right) GRF of Bcd to LacZ level from each constructs. Dots represent measured mRNA value and lines show a fit to Hill function. (**B**) From these data, we computed the Hill coefficient (nH), which reflects the shape of the curve, and generalised Michaelis-

*Figure 2 continued on next page*

*Figure 2 continued*

Menten constant (gMM) which reflects the location of the expression boundary along the anterior/posterior axis. The mRNA expression level at maximum (Emax) was measured for subsets of constructs using a co-stain method (*Wunderlich et al., 2014*). These quantities are plotted against the number of Bcd binding sites in each construct. (C) (left) Schematics of mutant hbP2 constructs where 1, 2, or 3 Bcd binding sites were removed at different positions. (center) Average expression profiles for each construct on the left. Colors are as indicated. Average is represented by the thick line, standard deviation is shown by the shaded area. We extracted both gMM and nH from these data by comparing Bcd input to the LacZ expression profile for each reporter construct. These analyses reveal that different Bcd binding sites have roughly equal contributions to the HbP2 GRF.
DOI: https://doi.org/10.7554/eLife.41266.004

The following figure supplement is available for figure 2:

**Figure supplement 1.** Residual expression remains even after removing all six canonical Bcd TFBS, and also two additional predicted TFBSs.
DOI: https://doi.org/10.7554/eLife.41266.005

---

sites contribute roughly equally to GRF shape, despite variations in affinity and position, which was also observed in other work (*Smith, 2015*).

## Bcd and other transcription factors contribute to expression driven by HbP2

HbP2 contains predicted binding motifs for TFs other than Bcd, including Hb itself and the pioneer factor Zelda (*Figure 3A*). To test if these other binding sites contribute to the WTHbP2 GRF, we created a synthetic HbP2 enhancer (synHbP2). This sequence maintains the endogenous Bcd binding sites in place but replaces the intervening sequences with computationally designed 'neutral sequences'; these neutral spacers are devoid of motifs for 11 TFs known to regulate *even-skipped* that are expressed in the blastoderm (Bcd, Cad, dStat, D, Gt, Hb, Kni, Kr, Nub, Tll, Zld) (*Estrada et al., 2016a*). We chose this limited list because designing sequences lacking motifs for all 37 TFs expressed in the blastoderm is technically intractable and often results in highly repetitive sequences. Bcd is a homeodomain protein and these are sensitive to DNA shape dictated by sequence outside of the core binding motif (*Dror et al., 2014*). We therefore made constructs with various amounts of endogenous sequence flanking each Bcd site. Robust expression requires 7 bp of endogenous sequence flanking each Bcd binding site; using only the 7 bp core Bcd binding motif, or motifs flanked by 2 bp did not drive robust expression (*Figure 3A*, right).

SynHbP2 drives expression comparable to WTHbP2, with a similar Hill coefficient but at an anteriorly shifted location in the embryo, corresponding to an increased gMM constant (*Figure 3B*). These results indicate that Bcd alone can mediate sharp expression but that other TFs, such as Hb and Zelda may affect the gMM constant and the location of the GRF along the AP axis. To further test the role of Hb and Zelda in mediating a sharp response from HbP2, we knocked down expression of Hb and Zelda in trans using RNAi (*Figure 3C*). Our RNAi strategy specifically knocks down the maternal contribution of these proteins (see Materials and Methods) (*Staller et al., 2013*). If SynHbP2's anterior shift is due to missing activity from Hb or Zelda, we expect similar anterior shifts in WTHbP2 expression when depleting these proteins in trans. Indeed, knock-down of hb and Zelda shifted the location of expression by 8% and 5% anteriorly along the AP axis without distinctive changes in the steepness of the response (nH(hbRNAi)=5.9 ± 0.6, nH(Zelda RNAi)=5.8 ± 0.9). Previous studies have noted an anterior shift in HbP2 expression when Zelda or Hb are perturbed, although the GRF was not quantitatively analysed (*Xu et al., 2014*). Our results confirm and quantitate this anterior shift, and reveal the differential influence of these perturbations on the steepness and location of the boundary.

To test how the GRF depends on the number of Bcd binding sites in SynHbP2, we deleted Bcd binding sites individually from the distal end, as we did for WTHbP2 (*Figure 3—figure supplement 1*). Removing Bcd binding sites from SynHbP2 shifted the boundary position further toward the anterior and reduced boundary steepness as observed with WTHbP2, demonstrating that the individual Bcd binding sites are functionally contributing to the GRF when in the synthetic background.

## Mediator and chromatin remodelers influence the shape of the HbP2 gene regulatory function.

Cooperativity can arise through a wide variety of potential molecular mechanisms, including chromatin remodeling or modification, or binding to a multi-valent cofactor such as Mediator. We therefore

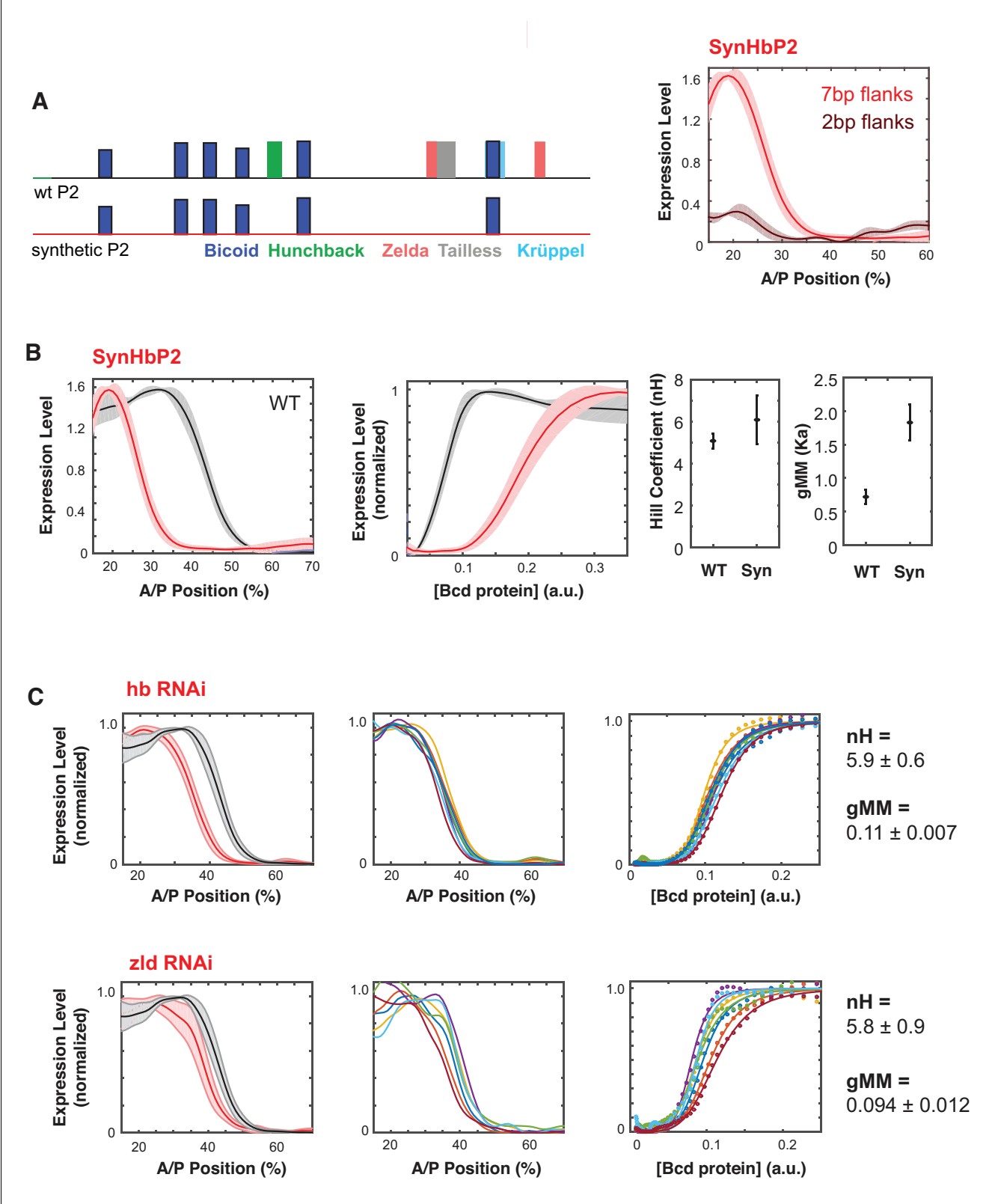

**Figure 3.** Synthetic HbP2 enhancer reveals the contribution of Bicoid and other transcription factors to the shape and location of expression. (**A**) Schematics of WTHbP2 illustrating the location and affinity of predicted binding sites for Bicoid (Bcd, dark blue), Hunchback (Hb, green), Zelda (Zld, red), Tailless (tll, grey) and Krüppel (Kr, light blue), and SynHbP2 where these other sites have been replaced by a computationally designed synthetic DNA backbone. In SynHbP2, Bcd binding sites are preserved in their native position and flanked by differing amounts of endogenous DNA. SynHbP2

*Figure 3 continued on next page*

*Figure 3 continued*

constructs containing Bcd binding sites without endogenous flanking sequences did not express (data not shown); adding back native sequence (+2 bp or +7 bp) flanking Bicoid TFBS restored expression (right). (B) SynHbP2 expresses in an anteriorly shifted position, but with a comparable shape. On the left, we compare the expression driven by WTHbP2 to expression driven by SynHbP2 with 7 bp flanks (hereafter SynHbP2) along the anterior posterior axis. On the right, we convert the data into the Bcd/LacZ GRF and extract nH and gMM as previously described. (C) We depleted Hunchback and Zelda from blastoderm embryos using RNAi to test their role in regulating WTHbP2. We show the average LacZ expression profiles from RNAi treated embryos (left column, thick line is the average over eight embryos, shaded area indicates standard deviation), the Bcd/LacZ GRF (center column), and the fit of the GRF to a Hill function to extract nH and gMM (right column). For reference, nH and gMM for WTHbP2 are 5.2 ± 0.3 and 0.072 ± 0.011, respectively.

DOI: https://doi.org/10.7554/eLife.41266.006

The following figure supplement is available for figure 3:

**Figure supplement 1.** Removing Bcd TFBS from SynHbP2 reporter construct.

DOI: https://doi.org/10.7554/eLife.41266.007

perturbed the expression of 12 transcriptional cofactors and known (or potential) interactors of Bcd using RNAi, and measured the effect on the WTHbP2 GRF (*Table 1*, *Table 2*). We knocked down the maternal expression of these factors using shRNAs available through TRiP (https://fgr.hms.harvard.edu/fly-in-vivo-rnai), and using the Gal4/UAS system (*Staller et al., 2013*). We observed significant effects for 5 of 12 target genes, see Materials and Methods and described below. Note that negative results in this experiment are not necessarily informative, as RNAi lines may not result in sufficient knockdown to observe a phenotype.

We hypothesized that chromatin remodelers might modulate the accessibility of regulatory DNA, and therefore influence the sensitivity of the WTHbP2 GRF to Bcd concentration and thus to the location of expression along the AP axis. For example, Creb Binding Protein (CBP) a histone acetyl transferase and the Drosophila ortholog of p300 (*Fu et al., 2004*), is known to destabilize chromatin and facilitate chromatin opening (*Chan and La Thangue, 2001*). CBP is also reported to be an enhancer-dependent co-activator of Bcd, facilitating its activation in S2 cells (*Fu et al., 2004*). Indeed, we found that knocking down CBP or HDAC6, both proteins involved in histone acetylation, results in a distinctive shift in location of expression toward the anterior (34 ± 2.5 %AP, 40 ± 1.5%AP respectively) without much change to the Hill coefficient (nH = 4.5 ± 0.53 and nH = 6.22 ± 0.03 respectively) (*Figure 4*, *Table 1*).

We also hypothesized that multi-valent cofactors, such as Mediator, might influence the shape of the WTHbP2 GRF by facilitating interactions between Bcd molecules. We found that knocking down

**Table 1.** List of shRNA lines

| Target gene name | TRiP ID | # of embryos |
|---|---|---|
| Nejire (CBP) | HMS01570 | 8 |
| Bin1/SAP18 | GL00127 | 8 |
| Sin3A | HMS00359 | 13 |
| HDAC6 | HMS00077 | 2 |
| Mediator complex subunit 1 | HMS01139 | 9 |
| Mediator complex subunit 11 | HMS01094 | 7 |
| Mediator complex subunit 14 | HMS01049 | 5 |
| Mediator complex subunit 20 | HMS01051 | 12 |
| Mediator complex subunit 22 | HMS01047 | 8 |
| Mediator complex subunit 27 | HMS01050 | 4 |
| Mediator complex subunit 28 | HMS00458 | 5 |
| Mediator complex subunit 7 | HMS01140 | 5 |
| Vielfaltig (Zld) | HMS02441 | 7 |
| hunchback | GL01321 | 11 |

DOI: https://doi.org/10.7554/eLife.41266.008

**Table 2.** Number of Embryos Imaged

| Reporter contructs | # of embryos |
|---|---|
| Endogeous hb | 6 |
| WThbP2 | 6 |
| WThbP2 (5TFBS) | 10 |
| WThbP2 (4TFBS) | 10 |
| WThbP2 (3TFBS) | 7 |
| WThbP2 (2TFBS) | 12 |
| WThbP2 (1TFBS) | 10 |
| WThbP2 (Δ1b) | 11 |
| WThbP2 (Δ2b) | 13 |
| WThbP2 (Δ3b) | 13 |
| SynHbP2 (2 bp) | 4 |
| SynHbP2 (7 bp) | 15 |
| SynHbP2 (12 bp) | 10 |
| SynHbP2 (12 bp, 4TFBS) | 15 |

DOI: https://doi.org/10.7554/eLife.41266.016

Mediator subunits had three types of effects on the WTHbP2 GRF. First, some subunits, such as MED14, reduce the overall expression level. Second, some subunits, such as MED20 and MED22, do not affect the Hill coefficient, but do affect the gMM constant. Finally, some subunits, such as MED11 and MED27, decrease the Hill coefficient, but do not affect the gMM constant (*Figure 5*).

## Comparing expression driven by variants of WTHbP2 and SynHbP2 to theory indicates multiple molecular mechanisms underlie the GRF

Fitting GRFs to Hill functions is a convenient way to characterize the shape of the response, but doesn't give any molecular insights. We developed a theoretical framework, rooted in molecular biophysics, to explore how individual TFs contribute to the GRF (*Estrada et al., 2016b*). If energy is not being expended to regulate transcription, so that regulation may be assumed to take place at thermodynamic equilibrium, our theory allows for arbitrary forms of 'higher-order' cooperativity (HOC), through which the affinity of binding at a site can be influenced by binding at multiple other sites. Such HOC allows any form of information integration which can be accomplished at thermodynamic equilibrium to be accommodated in a model. The molecular mechanisms that give rise to such higher-order effects may include chromatin, nucleosomes or co-regulators like Mediator. To assess the shape and location of the GRF, we introduced two non-dimensional parameters, called 'steepness' and 'position', which correspond to the maximal derivative and the location of the maximal derivative, respectively, of the normalised GRF. These intrinsic measures of shape are more biophysically informative than the nH and gMM constant of a fitted Hill function.

Higher order effects have rarely been considered previously. An exception is a series of quantitative studies of regulation of the Drosophila even-skipped gene, which are notable for their predictive accuracy (*Janssens et al., 2006*; *Kim et al., 2013*). The model underlying these studies, described in detail in (*Reinitz et al., 2003*), allows for higher order interactions of multiple binding sites, assuming thermodynamic equilibrium. These higher order effects seem important for the accuracy of the model. However, the effects are treated in a phenomenological way, which makes it difficult to draw quantitative comparisons with the biophysical model that we use.

In our previous work, mRNA expression was treated as an average over different TF binding microstates (*Estrada et al., 2016b*). For the present paper, we introduced a more refined model in which RNA polymerase is explicitly recruited to the promoter. In effect, RNA polymerase was treated as another molecular entity which can bind to DNA at a single site. The rate of mRNA expression was then taken to be proportional to the probability of RNA polymerase being bound. This model more accurately represents the behavior of TFs, which include domains that bind specifically to DNA, as well as domains which interact directly or indirectly with RNA polymerase. Accordingly,

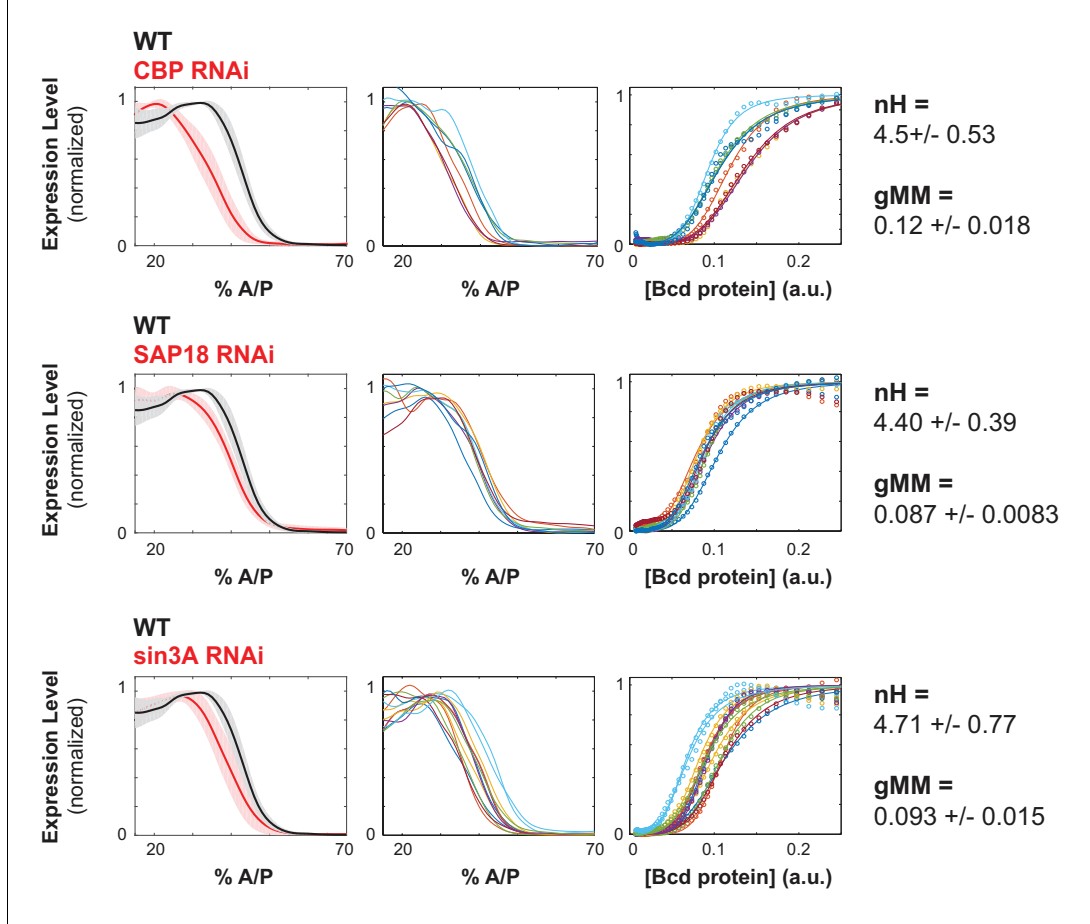

**Figure 4.** Chromatin remodelers and histone modifiers influence the WTHbP2 GRF. We knocked down the maternal contribution of multiple trans factors using RNAi in embryos harboring the WTHbP2 reporter. We stained for LacZ expression and present the average expression profile (left), the expression traces from individual embryos (center), and the Hill function fits of Bcd/LacZ GRFs from individual embryos (right). The number of embryos imaged is listed in *Table 2*. We present three proteins whose knockdown had a significant effect on the location of WTHbP2 expression. CBP is a histone acetylase that coactivates Bcd (*Fu et al., 2004*). SAP18 and Sin3A are cofactors of Bicoid that have histone deacetylase activity (*Singh et al., 2005*). For reference, nH and gMM for WTHbP2 are 5.2 ± 0.3 and 0.072 ± 0.011, respectively.
DOI: https://doi.org/10.7554/eLife.41266.009

higher-order cooperativities may arise between TF binding sites (TF-TF HOC) as well as between TF sites and RNA polymerase (TF-Pol HOC).

Here, we compare our model with various numbers of TF binding sites to our experimental data from deletions of WTHbP2 and SynHbP2 (*Figure 6*). In our model, we created GRFs by randomly choosing HOCs, for both TF-TF and TF-Pol, within a plausible range and determined the boundary of the position-steepness region occupied by any of such GRFs. This allows us to ask whether the position and steepness of the experimentally determined GRF is found within this biophysically plausible region. Note that this approach does not require fitting the GRF to experimental data. Instead, we mathematically determine the region of position and steepness in which the data should fall. We find that the WTHbP2 and SynHbP2 GRFs lie on the boundary of the position-steepness region, and that pairwise cooperativities, either TF-TF or TF-Pol, are inadequate to explain the data. Moreover, the deletion constructs yield GRFs whose steepness is higher than the model can accommodate, especially for low numbers of TF binding sites.

These discrepancies between our model and experiments indicate that additional features not included in our model must contribute to the steepness of the HbP2 GRF. For example, there may be cryptic Bcd binding sites that contribute to HbP2 expression. Alternatively, because our model accommodates all thermodynamically feasible mechanisms for a single transcriptional activator,

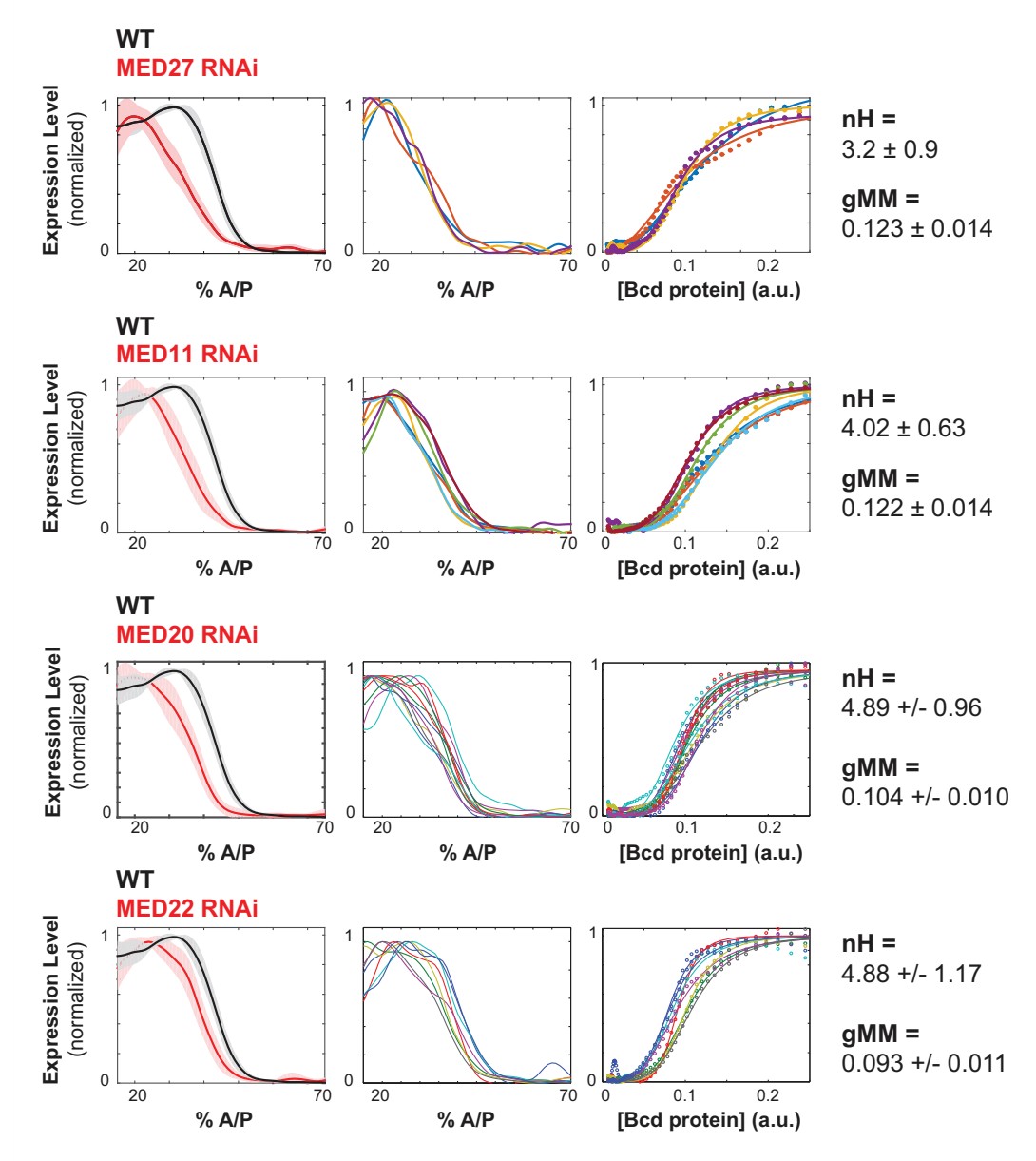

**Figure 5.** Mediator subunits affect the shape of the WTHbP2 GRF. Mediator subunits were knocked down using RNAi (see Materials and Methods). The number of embryos imaged is listed in *Table 2*. (Left column) Average gene expression profiles from hbP2-reporter construct in embryos with respective trans-factors depleted using RNAi; bold line is the average, shadow is the standard deviation. (Middle column) Individual gene expression profiles from each embryo; each color represents a separate embryo. (Right column) Bcd/LacZ mRNA GRF fit to the Hill function. Hill coefficient (nH) reflects the shape of the curve, generalized Michaelis Menten coefficient (gMM) reflects the location along the anterior posterior axis. For reference, nH and gMM for WTHbP2 are 5.2 ± 0.3 and 0.072 ± 0.011, respectively.

DOI: https://doi.org/10.7554/eLife.41266.010

additional regulators may play a crucial role. However, we attempted to control for this possibility by creating SynHbP2, which consists of only Bcd binding sites and still generates expression patterns with position and steepness beyond the limits of our model. We are left to consider whether non-equilibrium models requiring energy dissipation are central to generating the HbP2 GRF. We address these various possibilities further in the Discussion.

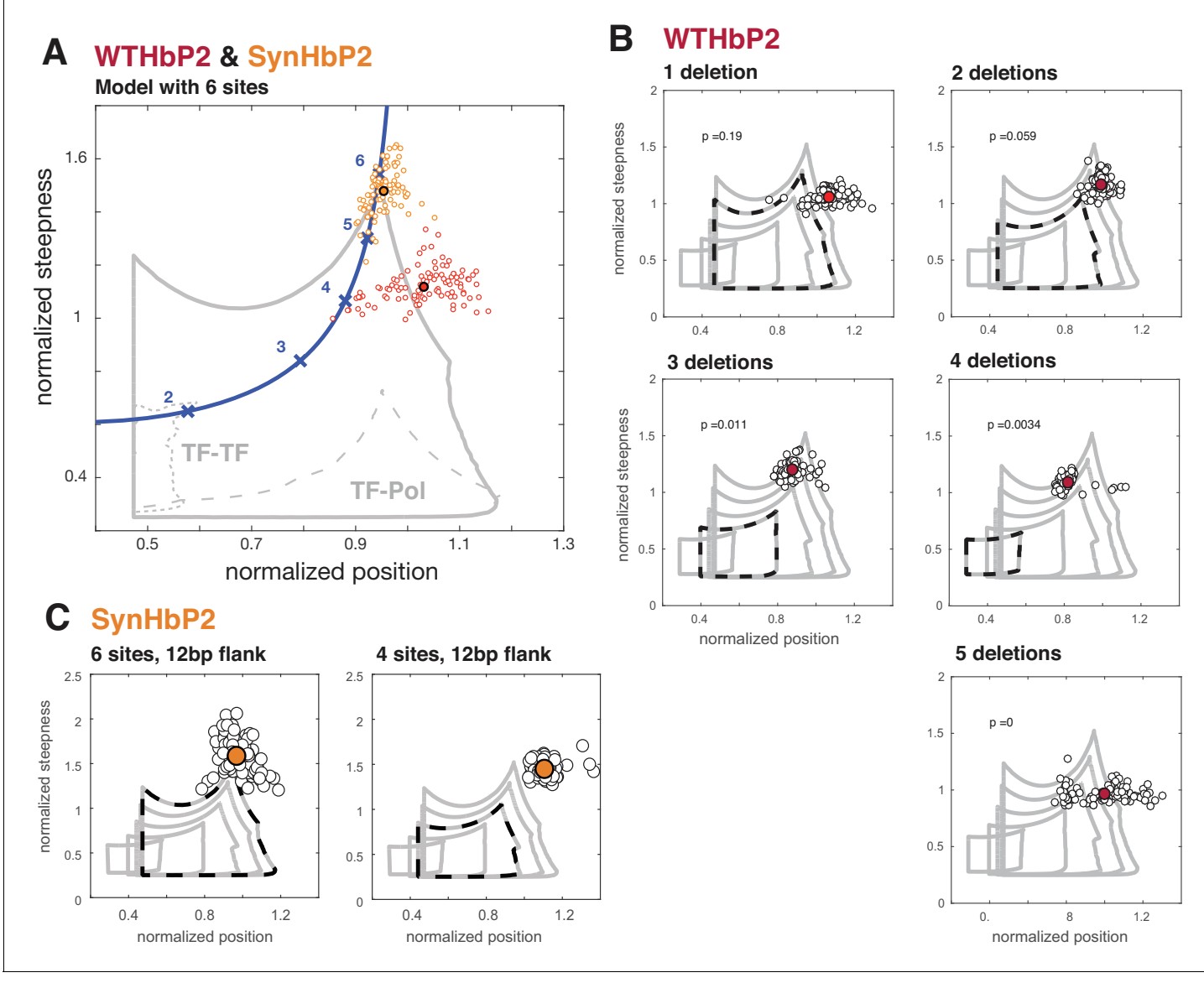

**Figure 6.** Deleting binding sites from WTHbP2 and SynHbP2 reveals discrepancies between experimentally measured GRFs and a model at thermodynamic equilibrium. (**A**) Using an updated version of the mathematical model in Estrada et al., we calculated the boundaries of the steepness and position for GRFs that can be generated by six transcription factor binding sites, either including higher-order cooperativities (solid grey line), pairwise TF-TF cooperativities (dashed grey line, lower left) or pairwise TF-Pol cooperativities (dashed grey line, lower right). For comparison, the Hill line, consisting of the position and steepness points for varying values of the Hill coefficient, nH, is plotted in magenta with the points corresponding to integer values of nH marked by crosses. Experimental data from WTHbP2 (individual embryos are shown as small red circles; the average is shown as a largered disc with a black surround) and SynHbP2 (individual embryos are shown as small orange circles; the average is shown as a large orange disc with a black surround). (**B**) We plotted the data from WTHbP2 variants where Bcd sites have been removed, described in *Figure 2*, in terms of steepness and position and compared it to the model with corresponding numbers of TF binding sites. The boundaries of the model for 2, 3, 4, 5 and 6 sites are shown in nested solid grey lines. The model corresponding to the number of remaining Bcd binding sites is shown in dashed black. Individual embryos are shown as small black circles; the average is shown as a red disc. P-values correspond to the probability of finding the GRFs inside the thermodynamic equilibrium region (see Materials and methods). (**C**) Data from two variants of SynHbP2 are plotted as in B, except the average is shown as an orange disc. In these constructs, the Bcd binding sites are flanked by 12 bp of endogenous sequence rather than 7 bp, as is shown in panel A. See supplementary data for the construct schematics and sequence details.

DOI: https://doi.org/10.7554/eLife.41266.011

The following figure supplements are available for figure 6:

**Figure supplement 1.** Deleting binding sites and comparing against model with extra sites.

DOI: https://doi.org/10.7554/eLife.41266.012

*Figure 6 continued on next page*

*Figure 6 continued*

**Figure supplement 2.** Procedure for obtaining normalized position (P) and normalized steepness (S) to compare the experimental data to the boundaries of mathematical GRFs calculated using equilibrium modeling.
DOI: https://doi.org/10.7554/eLife.41266.013
**Figure supplement 3.** Graph for gene regulation model.
DOI: https://doi.org/10.7554/eLife.41266.014
**Figure supplement 4.** Boundary stabilization.
DOI: https://doi.org/10.7554/eLife.41266.015

## Discussion

In 1989, Driever and Nusslein-Volhard proposed that Bcd binds cooperatively to the HbP2 enhancer to generate a sharp response from a graded input (*Driever et al., 1989*). Fitting the sigmoidal output of an HbP2 reporter construct to a Hill function revealed a Hill coefficient of 5 to 6, and this was interpreted to mean that Bcd bound cooperatively to HbP2, resulting in a GRF where *n* binding sites give rise to a Hill coefficient of *n*; this idea has been persistent in the Drosophila literature (*Gregor et al., 2007*; *He et al., 2010a*; *He et al., 2011*; *Lopes et al., 2012*; *Xu et al., 2015*). Here we test this model directly by making mutations in the HbP2 enhancer and measuring the consequences quantitatively. We first demonstrate that predicted relationship between binding site number and Hill coefficient does not hold. Then, we put forward data to support an alternative model where Bcd, other TFs and co-factors such as Mediator and CBP work together to shape the GRF through two largely separable properties - the steepness and position of the response. Finally, by comparing our results to theory, we propose that either unrecognized Bcd binding sites contribute to HbP2 expression, or that activation by a single TF at thermodynamic equilibrium may not be adequate to explain the response of HbP2, even when accounting for higher order cooperativities between TFs.

### Experimental distinctions between our work and previous studies

Our results agree qualitatively with previous studies, including those by Driever and Nusslein-Volhard. Upon removal of Bcd binding sites from HbP2, the expression boundary shifts toward the anterior and the sharpness decreases. However, our constructs do differ from that previous study in some important respects. Driever and Nusslein-Volhard removed Bcd TFBS by truncating WTHbP2 to shorter pieces. This strategy not only removes Bcd TFBS but also other neighboring sequences and it changes the distance of binding sites relative to the promoter. In contrast, our series of HbP2 variants preserve enhancer length, binding site positions and inter-motif sequences. All of our reporters used the native Hb promoter (*Perry et al., 2012*) rather than using the HSP 70 promoter (*Driever et al., 1989*) or a smaller fragment of the Hb promoter (+1 to+107) (*Driever et al., 1989*). But most significantly, we used quantitative imaging methods to measure the output of our HbP2 variants; this allows us to interpret our data differently. We find that the changes to the GRF upon binding site removal do not conform to the simple model where *n* binding sites can produce a GRF with a Hill coefficient of *n*. Instead, the Hill coefficient decreases by 0.5 with each site removed (rather than by one as would be predicted).

Our work also differs from recent quantitative studies on the Bcd/Hb GRF. In (*Gregor et al., 2007*; *He et al., 2010a*; *Lopes et al., 2012*), the authors measured the endogenous Hb protein expression profile, whereas we measured mRNA from an HbP2 reporter construct. In (*He et al., 2011*), the authors compared Bcd to endogenous hb mRNA expression using in situ hybridization. Importantly, all of these studies, including ours, largely agree, measuring Hill coefficients between 5 and 6. However endogenous hb mRNA (and endogenous Hb protein) cannot be directly interpreted as the output of the HbP2 GRF because it is difficult to exclude outputs from other hb enhancers that are known to contribute to anterior hb gene expression (*Perry et al., 2012*). For the purpose of deciphering the molecular details of an enhancer, our reporter designs and assays provide the most direct experimental tests. Lastly, we note that a recent quantitative fluorescence in situ hybridization study showed evidence of ~6 Bicoid molecules bound to the hb locus of earlier stage embryos (*Xu et al., 2015*). Our results suggest that either more than 6 Bcd molecules are interacting on HbP2, or energy is being spent to maintain the system away from equilibrium.

Multiple studies have focused on identifying interactions between Bcd molecules that could help explain cooperative binding. The studies used synthetic DNA sequences and mutated proteins in cell culture assays, and came to varied conclusions about the role of distance and clustering between Bcd binding sites (*Driever et al., 1989*; *Ochoa-Espinosa et al., 2005*). In our study using the endogenous enhancer in embryos, the arrangement, position, orientation, clustering and affinity of Bcd binding sites were not distinctive features for the degree of cooperativity. However, the number of perturbations we did to explore these parameters is limited, and it remains possible that they contribute partially to the degree of sharpness in the Bcd/HbP2 GRF.

## Flanking sequence influences bcd binding site efficacy

Bcd belongs to the homeodomain protein family, which recognize a common -TAAT- motif. While some homeodomain proteins have a degenerate specificity (*Berger et al., 2008*; *Christensen et al., 2012*; *Noyes et al., 2008a*), Bcd binds to the specific TAATCC motif (*Bergman et al., 2005*; *Noyes et al., 2008a*, *Noyes et al., 2008b*). Bcd has some unique structural features that allow this specificity, including recognition of -CC by direct contact with K50/R54 (*Baird-Titus et al., 2006*), and recognition of a weak binding motif (-TAAGCT-, X1) by the 'recognition helix' that harbors R54 (*Adhikary et al., 2017*; *Dave et al., 2000*; *Zhao et al., 2000*). This degree of specific binding makes it reasonable to assume that the core motif would be sufficient for Bcd binding and activity in vivo; however, here we show that including 7 bp of endogenous flanking sequence around each core Bcd motif dramatically increases the activity of a synthetic HbP2 enhancer. When we increased the flanking sequence to 12 bp, we didn't observe any additional effect. These flanking sequences do not have any obvious similarity to one another, though the number we have is far too small to analyze bioinformatically. We hypothesize that the flanks form a specific DNA shape that is favorable for Bcd recognition, as has been shown for other homeodomain proteins (*Dror et al., 2014*).

## Additional bcd binding sites outside HbP2

Even after all six known binding sites are removed from WTHbP2, we still observe gene expression with a sigmoidal, non-linear response. WTHbP2 does not express at all when Bcd is removed by RNAi (data not shown), indicating that some Bcd responsive element remains even after removing the canonical six sites. In our constructs, WTHbP2 is conjugated with its native hb promoter which is ~500 bp in length. The WTHbP2 sequence contains one additional predicted Bcd binding site in addition to the canonical six which overlaps with a weak binding site, and the native hb promoter also contains an additional predicted Bcd binding site. These two sites are obvious candidates for the residual Bcd responsive activity. We deleted these two additional sites and measured expression; expression is lowered further but still detectable (see *Figure 2—figure supplement 1*). We also calculated the limits of steepness and position assuming 7 or 8 TFBS and compared the boundaries to the experimental data from our TFBS deletion series. As expected, including 1 or two additional sites broadened the boundary of the steepness/position space. However, experimental data still often fall on the edge of the boundary and at lower number of TFBS, experimental data is outside the the boundary (see *Figure 6—figure supplement 1*). Because of the complexity of the native WTHbP2 enhancer sequence, we believe that synthetic constructs with defined binding site content such as the one we present here will be an important tool for dissecting the role of individual TF binding sites in dictating a response. In future studies, coupling our SynHbP2 to a synthetic promoter could also prove to be useful. We also note that promoters and enhancers have similar abilities to recruit PolII (*Henriques et al., 2018*), and especially in systems like ours where the enhancer is immediately adjacent to the promoter, may need to be considered as a single functional unit.

Importantly, our conclusion that pairwise cooperative binding of Bcd is inadequate to produce the observed expression from HbP2 is unaffected by how many Bcd sites there are (*Estrada et al., 2016b*). However, when considering the role of higher order cooperativity, additional binding sites allow the theory and experiment to be within range of one another. We therefore conclude that future efforts to understand the mechanism of cooperativity for WTHbP2 should focus on how higher order cooperativity can be achieved, or non-equilibrium mechanisms.

## Modulating the position of expression through effective concentration of Bcd

Multiple perturbations shifted the anterior/posterior location of expression driven by WTHbP2, reflecting a change in the sensitivity to Bcd concentration. We hypothesize that these perturbations all influence the accessibility of WTHbP2, and thus the ability of Bcd to bind. For example, removing binding sites for the pioneer factor Zelda may result in decreased accessibility of WTHbP2, a requirement for higher levels of Bcd to activate expression, and an anterior shift in expression. Similarly, Hb may also facilitate Bcd binding through collaborative cooperativity, and CBP may be necessary to modify nucleosomes. This hypothesis is consistent with previous studies where modulating Bcd concentrations influences the location of expression without altering the steepness (*Liu et al., 2013*; *Struhl et al., 1989*).

## Cofactors shape the gene regulatory function

Because of their central role in regulating transcription, TFs have been the primary focus of research to determine the molecular mechanisms underlying the shape of gene regulatory functions (*Janssens et al., 2006*; *Junion et al., 2012*; *Kazemian et al., 2010*; *Segal and Widom, 2009*; *Zinzen et al., 2009*). Specifically, pairwise cooperative binding between TFs has been widely proposed to underlie sharp responses to graded inputs (*He et al., 2010b*; *Segal et al., 2008*), though a series of models from the Reinitz group allow for higher-order cooperativity between TFs and other components (*Reinitz et al., 2003*; *Janssens et al., 2006*; *Kim et al., 2013*). In the scenario where cooperative binding between TFs dominates, the many cofactors crucial for eukaryotic gene regulation serve as 'cogs in the machine' to relay decisions from TFs to the basal transcriptional machinery.

Our work calls the TF-centric view into question using both theory and experiment. We show that theory grounded in physics must include higher order cooperativity between TFs to generate GRFs with sufficient steepness and position to compare to in vivo measurements; this assumes that the regulatory mechanism is operating at thermodynamic equilibrium, which may still be inadequate to capture experimental data, indicating that energy dissipation may be required. Cofactors are prime candidates for implementing higher-order cooperativity or energy dissipation through binding directly to multiple TFs or by facilitating their binding through indirect processes such as chromatin remodeling or modification. Indeed, we present experimental evidence that perturbing some cofactors changes the WTHbP2 GRF by altering either the anterior/posterior position or shape of expression. A role for cofactors in shaping the GRF is consistent with in vitro studies where purified Bcd protein binds to HbP2 but yields a Hill coefficient of only 3–4 (*Ma et al., 1996*). In independent work, we have shown that allosteric interactions arising from dynamically changing conformations can give rise to higher-order cooperativities (Biddle et al., submitted) and it is conceivable that cofactors may employ this mechanism at HbP2. We hope to investigate this in future studies. Together, these results argue that cofactors do not simply execute the decisions made by TFs; they collaborate with TFs to shape the quantitative features of the gene regulatory function.

Cooperativity between TFs mediated by cofactors may help to explain the rapid evolution of regulatory DNA. Because cooperative binding requires direct TF-TF contact, it should be a strong constraint on regulatory evolution. However, the number, position and affinity of WTHbP2 binding sites varies over evolutionary time. We speculate that cooperativity through cofactors may enable rapid evolution of regulatory sequence as motifs at disparate locations can still be integrated into the output of the enhancer.

## Broader implications

The shape of a gene regulatory function reflects the underlying molecular mechanisms of transcription. Coupling quantitative measurements to mechanistically meaningful biophysical models is a powerful strategy to decipher the individual role of such mechanisms, and how they collaborate to influence gene expression. This strategy has been enormously successful in bacteria (*Belliveau et al., 2018*; *Landman et al., 2017*; *Razo-Mejia et al., 2018*), and the toolbox for applying this strategy in animals is rapidly growing. Drosophila embryos are becoming a flagship system for this strategy due to key strengths developed by the community over time: the transcriptional network is well studied (*Levine, 2008*; *Nüsslein-Volhard and Wieschaus, 1980*), high resolution imaging of mRNA and protein is tractable (*Fowlkes et al., 2008*; *Gregor et al., 2014*; *Pisarev et al.,*

2009), and genetic and optogenetic perturbations are possible (*Huang et al., 2017*). Controlled synthetic systems, like the one we develop here and others that are emerging (*Crocker et al., 2017*), will be another crucial tool to harness the molecular complexity of animal transcription. As we show, coupling measurements to models has the power to reveal fundamental features of transcription that differ between animals and bacteria. As models improve, they will be a valuable tool in predicting the activity of animal enhancers and their variants and in engineering enhancers with desired activities, both of which are central goals of precision medicine.

# Materials and methods

## Key resources table

| Reagent type (species) or resource | Designation | Source or reference | Identifiers | Additional information |
|---|---|---|---|---|
| Strain, strain background (*Drosophila melanogaster*) | Nejire (CBP) RNAi | Bloomington Drosophila Stock Center | BDSC Cat# 36682, RRID:BDSC_36682 | |
| Strain, strain background (*D. melanogaster*) | Bin1/SAP18 RNAi | Bloomington Drosophila Stock Center | BDSC Cat# 36781, RRID:BDSC_36781 | |
| Strain, strain background (*D. melanogaster*) | Sin3A RNAi | Bloomington Drosophila Stock Center | BDSC Cat# 32368, RRID:BDSC_32368 | |
| Strain, strain background (*D. melanogaster*) | HDAC6 RNAi | Bloomington Drosophila Stock Center | BDSC Cat# 34072, RRID:BDSC_34072 | |
| Strain, strain background (*D. melanogaster*) | MED1 RNAi | Bloomington Drosophila Stock Center | BDSC Cat# 34662, RRID:BDSC_34662 | |
| Strain, strain background (*D. melanogaster*) | MED11 RNAi | Bloomington Drosophila Stock Center | BDSC Cat# 34083, RRID:BDSC_34083 | |
| Strain, strain background (*D. melanogaster*) | MED14 RNAi | Bloomington Drosophila Stock Center | BDSC Cat# 34575, RRID:BDSC_34575 | |
| Strain, strain background (*D. melanogaster*) | MED20 RNAi | Bloomington Drosophila Stock Center | BDSC Cat# 34577, RRID:BDSC_34577 | |
| Strain, strain background (*D. melanogaster*) | MED22 RNAi | Bloomington Drosophila Stock Center | BDSC Cat# 34573, RRID:BDSC_34573 | |
| Strain, strain background (*D. melanogaster*) | MED27 RNAi | Bloomington Drosophila Stock Center | BDSC Cat# 34576, RRID:BDSC_34576 | |
| Strain, strain background (*D. melanogaster*) | MED28 RNAi | Bloomington Drosophila Stock Center | BDSC Cat# 32459, RRID:BDSC_32459 | |
| Strain, strain background (*D. melanogaster*) | MED7 RNAi | Bloomington Drosophila Stock Center | BDSC Cat# 34663, RRID:BDSC_34663 | |
| Strain, strain background (*D. melanogaster*) | Vielfaltig (Zld) RNAi | Bloomington Drosophila Stock Center | BDSC Cat# 42016, RRID:BDSC_42016 | |
| Strain, strain background (*D. melanogaster*) | hunchback (hb) RNAi | Bloomington Drosophila Stock Center | BDSC Cat# 54478, RRID:BDSC_54478 | |

## Creating transgenic fly lines

The *D. mel* hb proximal enhancers (hbP2) and hb proximal promoter used were as previously defined (*Perry et al., 2012*). The proximal promoter was cloned with the following primers: Pfwd: cagtcagtcacgagtttgttac, Prev:cttggcggctctagacg. The HbP2 enhancer (−321 to +22), and HbP2 mutants were synthesized by Integrated DNA Technologies, Inc. For removing Bcd TFBS, the core Bcd binding sequence was modified as follows; A1, A2 and A3 (TAATC) were mutated to CCGAG; X1 and X2 (TAAGC) were mutated to CCGAG, and X3 (TCATC) was mutated to CCGAG (A1,A2,A3, X1,X2,X3 notations are from (*Driever et al., 1989*). The resulting sequences were checked for possible new binding motifs of AP patterning TFs (Bcd, Cad, dStat, D, Gt, Hb, Kni, Kr, Nub, Tll, Zld) using Patser (http://stormo.wustl.edu/software.html) with a p value of 0.001; the PWMs we used for these TFs (*Voss et al., 2011*; *Noyes et al., 2008a*) can be downloaded from Figshare (https://doi.org/10.6084/m9.figshare.8235587.v1).[2] Gibson assembly was used to replace the eve basal promoter of pBΦY with the hb proximal promoter. All other elements such as the LacZ reporter, Amp and mini-white marker genes, and an attB site for site-specific integration remained identical (*Groth et al., 2004*; *Hare et al., 2008*). Constructs were injected by Genetic Services and BestGene Inc, into the attP2 landing site. Flies were made homozygous using the mini-white marker.

## Creating synthetic enhancer constructs

To create SynHbP2, SiteOut was used to remove motifs of interest (*Estrada et al., 2016a*). First the sequence of WThbP2 was scrambled to remove known motifs of TFs (Bcd, Cad, dStat, D, Gt, Hb, Kni, Kr, Nub, Tll, Zld) involved in AP patterning of the blastoderm embryo. Then Bcd TFBS were restored to their native locations and the resultant sequence checked for any newly created motifs around the restored Bcd TFBS using PATSER (http://stormo.wustl.edu/software.html). The binding motifs are from FlyFactorSurvey (http://pgfe.umassmed.edu/ffs/; *Noyes et al., 2008b*), and a pseudocount of 0.1 and a GC (guanine and cytosine) content of 0.406 when generating position weight matrices from these count matrices was used. Full sequences for the resulting synthetic enhancers can be downloaded from Figshare (https://doi.org/10.6084/m9.figshare.8235614.v1). [3]

## RNAi screening

To measure reporter expression in RNAi backgrounds, we followed the protocol developed in (*Staller et al., 2013*). Briefly, virgin females with a maternal-tubulin-Gal4 driver were crossed to males with a UAS-short hairpin RNA (shRNA) construct. We then collected virgin female offspring and crossed them to males bearing hbP2-reporter constructs. The resulting embryos were collected and fixed for in situ hybridization. The UAS-shRNA line for each gene can be found in supplementary information with Transgenic RNAi Research Project [TRiP] number.

## In situ Hybridization

In situ hybridizations were performed as previously described (*Luengo Hendriks et al., 2006*). Flies were raised at 25°C and embryos aged 0- to 4-hr-old were collected and fixed. Embryos were incubated at 56°C for 2 days with 2,4-dinitrophenyl (DNP)-labeled probes for *lacZ* and *hkb* and digoxigenin (DIG)-labeled probes for *ftz*. For the expression level comparison among transgenic reporter lines, hkb was costained for normalization as described in (*Wunderlich et al., 2014*). Probes were sequentially detected with anti-DIG HRP (horseradish peroxidase) antibody (Roche) plus coumarin-tyramide color reaction (PerkinElmer) and anti-DNP HRP (PerkinElmer) antibody plus Cy3-tyramide color reaction (PerkinElmer). Embryos were treated with RNase A and then nuclei stained with Sytox Green (Life Technologies). Embryos were mounted in DePex (Electron Microscopy Sciences), using a bridge of #1 coverslips to preserve embryo morphology.

## Image acquisition, Processing, and Analysis

A two-photon laser scanning microscope (Zeiss LSM 710) with a plan-apochromat 20 × 0.8 NA objective was used to acquired z-stacks of each embryo. Each stack was converted into a Point-Cloud; a text file that includes the location and levels of gene expression for each nucleus (*Luengo Hendriks et al., 2006*). Embryos were imaged in the early blastoderm stage (0%–4% membrane invagination). *LacZ* levels were normalised to the 95% quantile of *hkb* expression in the posterior 10% of each embryo. (*Wunderlich et al., 2014*). Importantly *lacZ* levels were only compared

between other embryos stained within the same batch to avoid extraneous sources of noise in the normalization. Line traces of embryos were generated using the extractpattern command in the PointCloud toolbox (http://bdtnp.lbl.gov/Fly-Net/bioimaging.jsp?w=analysis). This divides the embryo into 16 strips along the AP axis of the embryo and, for each strip, calculates the mean expression level in 100 bins along the AP axis. Strips were extracted along the right lateral sides of the embryos and subtracted the minimum value along the axis to remove background noise.

## Estimating position and steepness from experimental data

For each embryo, the expression level driven by the corresponding reporter construct is measured as described above. The Bcd expression profile is estimated from a single sample of six embryos stained with Bcd antibody. This allows an estimate of a gene regulatory function (GRF) for each reporter construct, where expression level is expressed as a function of Bcd concentration. Normalized position (P) and normalized steepness (S) are estimated from an averaged GRF obtained by averaging over a random subsample of half the number of embryos in a particular condition. Given an averaged GRF, we estimate normalized steepness and position by: 1) Computing Bcd05: Bcd concentration for which expression level is half its maximum value 2) Computing the maximum derivative of the GRF (raw steepness: rS) and the Bcd concentration for which it is found (raw position: rP) 3) Normalizing the raw steepness and raw position using Bcd05: p=rP/Bcd05 and S = rS*Bcd05. 100 random subsamples per condition are taken, allowing to compute a distribution of possible positions and steepnesses for a given condition. Kernel Density Estimation is used to estimate a probability distribution for the particular reporter construct to exhibit a particular combination of steepness and position. A gaussian basis is used, where the bandwidth is estimated using *Silverman's rule of thumb* (*Silverman, 2018*), which is optimal, provided the underlying distribution is, in fact, Gaussian. A p-value for the probability of finding GRFs inside the thermodynamic equilibrium region is estimated by computing the proportion of the estimated kernel density that lies within the equilibrium boundary (*Figure 6—figure supplement 2*).

## Model of gene regulation

The biophysical approach to gene regulation used in (*Estrada et al., 2016b*) was introduced in (Ahsendorf, Wong, Eils, Gunawardena, '*A framework for modelling gene regulation which accommodates non-equilibrium mechanisms*', BMC Biology 12:102 2014). It allows for multiple molecular entities interacting with DNA, giving rise to 'microstates', corresponding to the various molecular patterns which can arise on DNA, and transitions between these microstates, corresponding, for instance, to binding and unbinding. This specifies in a graphical way the master equation of the underlying Markov process, from which the steady-state probabilities of the microstates can be calculated algebraically, without having to know the numerical values of the transition rates. If the steady-state is one of thermodynamic equilibrium, then the model can be parameterised by association constants, corresponding to ratios of on-rates to off-rates, along with the concentrations of the interacting molecular entities. The steady-state probabilities are identical to those given by equilibrium statistical mechanics for the grand canonical formalism. The parameterisation by association constants is more convenient for representing biochemical mechanisms than the conventional formulation in terms of free energies of microstates. The treatment accommodates any form of information integration that is achievable without energy expenditure.

The mathematical foundations for this approach were developed in previous work on the graph-based 'linear framework' (see *Estrada et al., 2016a* for references) and are summarised for the specific application to gene regulation in Section 1 of the Supplementary Information of (*Estrada et al., 2016b*). We provide sufficient information here to reproduce the results of the present paper, based on these previously published results. This general approach to gene regulation was applied to the Hb-Bcd system in (*Estrada et al., 2016a*), assuming a single activating transcription factor (TF) binding at any number of sites, with mRNA synthesis given by averaging over the steady-state probabilities of the microstates. Higher-order cooperativites (HOCs), defined as ratios of association constants, were introduced as non-dimensional parameters which measure information integration.

For the present paper, we followed the same approach with a slightly extended model, in which RNA polymerase was introduced as a molecular entity with a single binding site, corresponding to the transcription start site (*Figure 6—figure supplement 3*). Higher-order cooperativity can now

arise from subsets of TF sites acting on a TF site (TF-TF HOC) and between subsets of TF sites acting on RNA polymerase (TF-Pol HOC) (*Figure 6—figure supplement 3*, bottom inset). The mRNA synthesis rate is assumed to be proportional to the steady-state probability of RNA polymerase being bound and the steady-state amount of mRNA is then determined as the balance between synthesis and linear degradation (*Estrada et al., 2016a*, SI, Section 5). Using the rules described in (*Estrada et al., 2016b*, SI, Section 1), the normalised mRNA amount can be calculated algebraically in terms of the concentration of the TF and the HOCs, giving thereby a fully specified gene regulation function (GRF).

To measure the sharpness of this GRF, we follow the method in (*Estrada et al., 2016b*) by using 'steepness' (maximum of the derivative of the GRF) and 'position' (normalised concentration at which the maximum is achieved). To compare these biophysically defined GRFs to data, we do not fit them but, instead, we plot the region of position and steepness which is accessible, assuming that the HOC parameters are independently chosen as $10^u$, where u is drawn at random from the uniform distribution on $[-3, 3]$. This range seems plausible on biophysical grounds but the boundaries of the position-steepness regions stabilise rapidly as the parameter range is increased (*Estrada et al., 2016a*). Changing the range from $[-3, 3]$ to $[-4,4]$ changes the boundary only slightly, indicating that the boundary has nearly stabilized, and does not affect our results (*Figure 6—figure supplement 4*).

## Acknowledgements

The authors wish to acknowledge all members of the DePace and Gunawardena groups, for their helpful feedback on the work and manuscript. In particular, we wish to thank Ed Pym for his help with finalizing the manuscript, and Clarissa Scholes for her help in making the first HbP2 reporter constructs. We also wish to thank former DePace lab members Tara Lydiard-Martin and Max Staller for their technical help and insightful discussions in the early stages of this work.

## Additional information

### Funding

| Funder | Grant reference number | Author |
| --- | --- | --- |
| National Institutes of Health | 5K99HD073191-02 | Zeba Wunderlich |
| National Institutes of Health | R01GM122928 | Jeremy Gunawardena Angela H DePace |
| Giovanni Armenise-Harvard Foundation | | Angela H DePace |

The funders had no role in study design, data collection and interpretation, or the decision to submit the work for publication.

### Author contributions

Jeehae Park, Javier Estrada, Conceptualization, Investigation, Formal Analysis, Writing-original draft; Gemma Johnson, Ben J Vincent, Chiara Ricci-Tam, Investigation; Meghan DJ Bragdon, Anna Cha, Zeba Wunderlich, Data curation; Yekaterina Shulgina, Investigation, Formal Analysis; Jeremy Gunawardena, Angela H DePace, Conceptualization, Resources, Supervision, Funding acquisition, Project administration, Writing-review and editing

### Author ORCIDs

Javier Estrada (iD) https://orcid.org/0000-0003-3646-204X
Chiara Ricci-Tam (iD) https://orcid.org/0000-0002-6549-1838
Jeremy Gunawardena (iD) https://orcid.org/0000-0002-7280-1152
Angela H DePace (iD) https://orcid.org/0000-0001-5723-0438

### Decision letter and Author response
Decision letter https://doi.org/10.7554/eLife.41266.025

Author response https://doi.org/10.7554/eLife.41266.026

## Additional files

### Supplementary files
• Transparent reporting form
DOI: https://doi.org/10.7554/eLife.41266.017

### Data availability

All data generated or analyzed during this study are included in the manuscript or linked to at Figshare.

The following datasets were generated:

| Author(s) | Year | Dataset title | Dataset URL | Database and Identifier |
|---|---|---|---|---|
| Angela H DePace, Jeehae Park, Gemma Johnson, Chiara Ricci-Tam, Ben J Vincent, Meghan DJ Bragdon, Anna Cha, Zeba Wunderlich | 2019 | 3D dataset.zip | https://doi.org/10.6084/m9.figshare.8235491.v1 | figshare, 10.6084/m9.figshare.8235491 |
| Angela H DePace, Jeehae Park | 2019 | PWMs used for HbP2 sequence analysis | https://doi.org/10.6084/m9.figshare.8235587.v1 | figshare, 10.6084/m9.figshare.8235587 |
| Angela H DePace, Jeehae Park, Javier Estrada | 2019 | Synthetic Hunchback P2 Sequences | https://doi.org/10.6084/m9.figshare.8235614.v1 | figshare, 10.6084/m9.figshare.8235614 |

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
