## [Decision Letter]

Thank you for submitting your article "Dissecting the sharp response of a canonical developmental enhancer reveals multiple sources of cooperativity" for consideration by *eLife*. Your article has been reviewed by three peer reviewers, and the evaluation has been overseen by a guest Reviewing Editor and Patricia Wittkopp as the Senior Editor. The reviewers have opted to remain anonymous.

The reviewers have discussed the reviews with one another and the Reviewing Editor has drafted this decision to help you prepare a revised submission.

Summary:

In this manuscript, the authors have quantitatively assessed the in vivo activity of the *Drosophila* hunchback regulatory element controlled by Bicoid binding sites in the embryo. Simple cooperative binding models do not reproduce the effects of perturbations, suggesting that either the thermodynamic equilibrium assumption is invalid for these types of elements, or that the cooperativities are more complex, possibly involving the influence of cofactors, chromatin, or DNA shape. The authors test the influence of additional transcriptional regulators by cis-element mutations and RNAi knockdowns. Previous studies showed that Zld and Hb are involved in Bcd-dependent activation, and these effects are confirmed using shRNA interference of each gene. The perturbations also include disruption of the general transcriptional machinery as well (i.e. Mediator). The reviewers found worth in the quantitative assessments and modeling for the gene regulatory output function of the hunchback promoter. Although the main points were to rule out the simplest models, the work points to a productive path forward and provide valuable quantitative data for modeling.

Essential revisions:

In a number of places, the authors oversimplify current and past discussions about sources of cooperativity, function of cofactors, and neglect earlier efforts to incorporate some of the additional layers of interactions in cooperative models (e.g. Reinitz, (2003) Complexus; Janssens et al., 2006, Kim et al. (PLoS Genetics) 2013].).

Regarding the number of Bicoid sites and Hill coefficients, among the papers that the authors cited for the sentences above, only Gregor et al., 2007, mention the matching relationship between Hill coefficients (nH = 5) and the number of Bcd molecules (n=5, but not the number of sites, Gregor mentioned 7 binding sites in the paper). Furthermore, many previous studies including ones the authors cited mentioned that additional or unknown factors or mechanisms might be required to fully explain the measured Hill coefficients or transcriptional response of Hb promoters.

It would be important to understand how the PWMs used here for Bcd impact model sensitivity, selectivity, specificity. The quality of PWMs can significantly affect its accuracy in predicting TF binding and because HbP2 mutant sequences used in this study were made based on the PWMs, the test of predictive capability of the PWMs can strongly support the author's arguments in relation to the mutants.

The Hill Equation given in the subsection “The HbP2 reporter system”, is an incredibly simple model of regulation. This model assumes that Bcd is the *only* mechanism through which HbP2 is being activated. The model also assumes all sites are 'equal', without considering relative spacing, affinity, etc. Therefore, it is not surprising at all that the HbP2 variants tested do not adhere to this model framework. This limitation should be addressed in the Discussion.

A more complex model involves post-binding cooperative interactions with the basal machinery; details of this model must be presented. In the subsection “Comparing expression driven by variants of WTHbP2 and SynHbP2 to theory indicates multiple molecular mechanisms underlie the GRF”, the authors introduce a more refined model. However, the model itself (full description, equations, etc.) is not presented. This should be included in the Materials and methods section, as well as all parameter values used in the model and specifically which higher-order terms were included during parameter fitting. It is very hard to assess whether the expression patterns are "beyond the limits" of their model without this information.

---

## [Author Response]

Essential revisions:In a number of places, the authors oversimplify current and past discussions about sources of cooperativity, function of cofactors, and neglect earlier efforts to incorporate some of the additional layers of interactions in cooperative models (e.g. Reinitz, (2003) Complexus; Janssens et al., 2006, Kim et al. (PLoS Genetics) 2013].).

Thank you for pointing out this oversight. We have modified the manuscript to discuss these earlier modeling efforts (see Results section “Comparing expression driven by variants of WTHbP2 and SynHbP2 to theory indicates multiple mechanisms underlie the GRF”).

Regarding the number of Bicoid sites and Hill coefficients, among the papers that the authors cited for the sentences above, only Gregor et al., 2007, mention the matching relationship between Hill coefficients (nH = 5) and the number of Bcd molecules (n=5, but not the number of sites, Gregor mentioned 7 binding sites in the paper). Furthermore, many previous studies including ones the authors cited mentioned that additional or unknown factors or mechanisms might be required to fully explain the measured Hill coefficients or transcriptional response of Hb promoters.

We have tempered our claim that Hill coefficient = binding site number is commonly accepted, to indicate this important complexity in the literature (Introduction, fourth paragraph).

It would be important to understand how the PWMs used here for Bcd impact model sensitivity, selectivity, specificity. The quality of PWMs can significantly affect its accuracy in predicting TF binding and because HbP2 mutant sequences used in this study were made based on the PWMs, the test of predictive capability of the PWMs can strongly support the author's arguments in relation to the mutants.

The reviewers are right to point out that the quality of Position Weight Matrices (PWMs) impacts their ability to accurately predict TF binding. PWMs do not impact our models (which do not explicitly contain DNA sequence), but they do impact the design of the SynHbP2 construct. For our experiments on WTHbP2, we manipulated the footprinted Bcd binding sites, which does not require the use of PWMs. For SynHbP2, we used PWMs to predict binding sites for other TFs to screen against the creation of those sites in our neutral backbone. To explore the possibility that additional Bcd binding sites exist, outside of the footprinted sites, we also used a PWM for Bcd, and we now include a supplemental figure addressing that possibility. Note that it does not substantively alter our conclusions. We added full information on PWMs we used in the Materials and methods (subsection “Creating Transgenic Fly Lines”) with link to Figshare (https://doi.org/10.6084/m9.figshare.8235587.v1) .

The Hill Equation given in the subsection “The HbP2 reporter system”, is an incredibly simple model of regulation. This model assumes that Bcd is the only mechanism through which HbP2 is being activated. The model also assumes all sites are 'equal', without considering relative spacing, affinity, etc. Therefore, it is not surprising at all that the HbP2 variants tested do not adhere to this model framework. This limitation should be addressed in the Discussion.

We agree that the Hill equation is simple, but nonetheless it is a persistent tool in the literature, and therefore must be investigated as we do here. We discuss the simplicity of this model in the paper (*“*As noted above, the Hill function gives little biophysical insight but it offers a widelyused measure of shape. We will later interpret the shape of the GRF in terms of the mathematical theory that we introducedpreviously, from which we will draw more mechanistic conclusions.”), and do not claim that it is surprising that it doesn’t fit. Instead, we use the lack of agreement with the predictions of the Hill equation to justify more complex and mechanistically insightful models. We now reference other such mechanistic models, even though they were developed for other enhancers, in the Results section as described above.

In regards to the equivalence of sites, we tested this explicitly in the paper (Figure 2), and found that they were indeed roughly equal contributors. This is consistent with work from Thomas Gregor’s lab (Smith, 2015).

A more complex model involves post-binding cooperative interactions with the basal machinery; details of this model must be presented. In the subsection “Comparing expression driven by variants of WTHbP2 and SynHbP2 to theory indicates multiple molecular mechanisms underlie the GRF”, the authors introduce a more refined model. However, the model itself (full description, equations, etc.) is not presented. This should be included in the Materials and methods section, as well as all parameter values used in the model and specifically which higher-order terms were included during parameter fitting. It is very hard to assess whether the expression patterns are "beyond the limits" of their model without this information.

We absolutely agree that this information is necessary, and apologize that it was not included in the initial submission. We have revised the main text (see subsection “Comparing expression driven by variants of WTHbP2 and SynHbP2 to theory indicates multiple molecular mechanisms underlie the GRF”) and added a description to the Materials and methods, including a new supplementary figure describing the graph that we use, which indicates the notation, and describes the rules for calculating steady states (see Figure 6—figure supplement 3 and Figure 6—figure supplement 4). These rules are extensively described in the supplement to Estrada et al., 2016, so we do not reproduce them here. This information is sufficient to reconstruct our model, but does not delineate every equation and parameter. We are happy to elaborate in further detail if the reviewers deem it necessary.